Research on marine flexible biological target detection based on improved YOLOv8 algorithm

Tian Yu
Liu Yanwen
Lin Baohang
http://orcid.org/0000-0002-8424-1367 Li Peng lipeng@hrbeu.edu.cn
College of Intelligent Systems Science and Engineering, Harbin Engineering University , Harbin , China
Marsland Stephen
Electronic publication date: 2024 Aug 22
Publication date: 2024
Volume: 10
Electronic Location ID: e2271
Received 2023 Nov 1; Accepted 2024 Jul 29
Copyright: © 2024 Tian et al.
Copyright year: 2024
Copyright holder: Tian et al.
License: This is an open access article distributed under the terms of the Creative Commons Attribution License, which permits unrestricted use, distribution, reproduction and adaptation in any medium and for any purpose provided that it is properly attributed. For attribution, the original author(s), title, publication source (PeerJ Computer Science) and either DOI or URL of the article must be cited.
License URL: https://creativecommons.org/licenses/by/4.0/

Keywords: Marine flexible biological targets, Target detection, CLAHE, Improved YOLOv8

Funding: State Key Laboratory of Robotics Technology and Systems SKLRS-2023-KF-17 This work was supported by the State Key Laboratory of Robotics technology and Systems open fund (No. SKLRS-2023-KF-17). The funders had no role in study design, data collection and analysis, decision to publish, or preparation of the manuscript.

==============================
To address the challenge of suboptimal object detection outcomes stemming from the deformability of marine flexible biological entities, this study introduces an algorithm tailored for detecting marine flexible biological targets. Initially, we compiled a dataset comprising marine flexible biological subjects and developed a Contrast Limited Adaptive Histogram Equalization (CLAHE) algorithm, supplemented with a boundary detection enhancement module, to refine underwater image quality and accentuate the distinction between the images’ foregrounds and backgrounds. This enhancement mitigates the issue of foreground-background similarity encountered in detecting marine flexible biological entities. Moreover, the proposed adaptation incorporates a Deformable Convolutional Network (DCN) network module in lieu of the C2f module within the YOLOv8n algorithm framework, thereby augmenting the model’s proficiency in capturing geometric transformations and concentrating on pivotal areas. The Neck network module is enhanced with the RepBi-PAN architecture, bolstering its capability to amalgamate and emphasize essential characteristics of flexible biological targets. To advance the model’s feature information processing efficiency, we integrated the SimAM attention mechanism. Finally, to diminish the adverse effects of inferior-quality labels within the dataset, we advocate the use of WIoU (Wise-IoU) as a bounding box loss function, which serves to refine the anchor boxes’ quality assessment. Simulation experiments show that, in comparison to the conventional YOLOv8n algorithm, our method markedly elevates the precision of marine flexible biological target detection.

Introduction

With the rapid advancement of target detection algorithms, the detection accuracy for common tasks in simple scenarios, like pedestrian detection, now exceeds 90%. However, for targets with unique characteristics, like the flexible biological entities discussed herein, the algorithm’s performance remains unsatisfactory. Consequently, it is necessary to customize and improve the algorithm according to these unique characteristics to improve the detection results.

Observing flexible marine biology offers a valuable method for assessing marine biological diversity and forecasting environmental shifts. Recently, underwater target detection employing machine vision has gained traction in marine studies. Utilizing visual autonomous perception technology to precisely identify and quantify various marine species, including sea cucumbers and octopuses in aquatic farms, aids seafood cultivators in real-time monitoring of marine life growth and environmental alterations, while also minimizing human labor and avoiding high-risk tasks. Despite this potential, four significant challenges hinder effective detection of flexible biological entities in marine settings (Lin & Zhao, 2020):

1) Flexible marine biological objects exhibit considerable variability in shape and posture due to their soft, deformable bodies.

2) Detection backgrounds in marine environments are often complex and variable.

3) Marine organisms excel in camouflage and concealment compared to inanimate objects.

4) In underwater environments, light absorption, scattering, and water quality significantly affect underwater imagery, often resulting in color distortion, blurring, and detail loss.

Based on the preceding analysis, the challenges can be categorized into two primary areas (Chen et al., 2023a): underwater image processing and object detection algorithms. Underwater image processing encompasses methods for image enhancement and image restoration (Wang et al., 2019). Enhancement techniques include prior model-based approaches such as ULAP and IBLA (Song et al., 2018; Peng & Cosman, 2017), which rely on predetermined knowledge or statistical assumptions about the environment. However, these assumptions may not hold in complex scenarios. Conversely, non-physical model-based methods, like the CLAHE algorithm, utilize histogram equalization to improve image clarity. Another example is the Relative Global Histogram Stretching (RGHS) algorithm, which enhances shallow water images by adjusting contrast and color, thereby achieving superior image quality with reduced noise (Huang et al., 2018). Recently, deep learning-inspired methods involving convolutional neural network (CNN) and generative adversarial network (GAN) like Water-net, have emerged. These methods employ a network’s self-learning capability to conduct weighted fusion of input images, resulting in enhanced underwater imagery (Li et al., 2019).

To tackle underwater image restoration, He, Sun & Tang (2020) introduced the Dark Channel Prior (DCP) algorithm, leveraging prior knowledge for image restoration. Chiang’s team (Raveendran, Patil & Birajdar, 2021) applied DCP to underwater contexts and developed a restoration algorithm that incorporates wavelength compensation and deblurring. Beyond DCP, Wang, Zheng & Zheng (2017) presented the Adaptive Attenuation-Curve Prior (AACP) algorithm for similar purposes. In the realm of object detection, algorithms are bifurcated into traditional methods and those predicated on deep learning (Wei et al., 2023). Traditional approaches (Zou et al., 2023), such as Cascade+Haar, DP (Deformable Parts Model), and Selective Search (SIFT+SVM), are becoming outdated due to technological advancements and increasing data volumes. These methods often suffer from low computational efficiency due to the sliding window technique and are hampered by the manual intensity required for feature design and selection, affecting their accuracy, objectivity, robustness, and generalizability. Deep learning-based object detection algorithms are categorized into two types: a) two-stage algorithms, which separate detection into region proposal extraction and proposal classification and regression, with RCNN (Girshick et al., 2014), Fast region-based convolutional neural network (R-CNN) (Sabri & Li, 2021), Faster R-CNN (Wang & Xiao, 2023) and Mask R-CNN (Rashid et al., 2022) as prime examples; b) one-stage algorithms, which concurrently perform detection and classification, outputting class probabilities and object coordinates directly, exemplified by Cao et al.’s (2020) SSD and the YOLO series (Jiang et al., 2022) algorithms.

The performances of object detection algorithms, including those based on deep learning like YOLO and SSD (Chen, 2023; Dong et al., 2022), often diminishes with poor imaging quality and complex or varied detection backgrounds. Furthermore, the detection of flexible marine biological objects (Liu, Liu & Jiang, 2023), characterized by their soft bodies and significant shape changes, demands an algorithm with enhanced modeling capabilities for geometric transformations and a robust distinction between the foreground and background of an image.

Accordingly, this article introduces a flexible marine biological object detection algorithm using an enhanced YOLOv8. “Improvements of the CLAHE Underwater Image Enhancement Algorithm” discusses the incorporation of the CLAHE image enhancement algorithm with a boundary detection enhancement module to improve the underwater image dataset and delineate the boundaries between the image’s foregrounds and backgrounds. “The Structure of Yolov8n Model and Improvement Strategy” outlines the comprehensive framework of the object detection algorithm based on YOLOv8n, detailing improvements in the backbone, neck, and loss function, and integrating the attention mechanism SimAM. “Experiment Research and Result Analysis” describes the assembly of a flexible marine biological dataset, the establishment of an experimental environment, and the execution of ablation studies along with comparative analyses of different models. “Conclusions” concludes the article.

Improvements of the clahe underwater image enhancement algorithm

CLAHE (Ting et al., 2023) is an advanced image enhancement technique evolved from Adaptive Histogram Equalization (AHE). The fundamental concept of CLAHE is to partition the image into multiple segments (or tiles), for example, 8 × 8 = 64 tiles. Unlike AHE, where histogram equalization is applied to the entire image, CLAHE performs this operation individually on each tile. While AHE excels in enhancing local image contrast and delineating image boundaries, it falls short by intensifying image noise. To alleviate this problem, CLAHE imposes a contrast limitation on each tile to prevent the excessive noise amplification associated with AHE. The operational principle of CLAHE is outlined as follows:

Assuming that the sliding window size of the CLAHE algorithm is M×M, therefore the local mapping function is

(1) m(i)=255×CDF(i)M×M.

CDF(i) is the cumulative distribution function of the sliding window local histogram; the slope S of the local mapping function is

(2) S=d(m(i))di=Hist(i)×255M×M.

According to Formulas (1) and (2), by adjusting S, the height Hist(i) can be modified, which in turn adjusts the image contrast. If the maximum value of S is represented as Smax, the maximum value of the histogram is

(3) Hmax=Smax×M×M255.

To maintain a consistent overall area of the histogram, the part of the histogram that exceeds Hmax is truncated and evenly distributed within the entire histogram range. In practice, this truncation is achieved by setting a threshold T (instead of Hmax), which results in the entire image’s histogram increasing by L. Consequently,

(4) Hmax=T+L.

Finally, the improved histogram is

(5) Hist(i)={Hist(i)+L,Hist(i)<THmax,Hist≥T.

To summarize, by adjusting Smax, histograms with different maximum heights Hmax can be enhanced to varying degrees. To augment the algorithm’s capacity for delineating foreground and background boundaries in addition to enhancing underwater images, a Boundary Detection Enhancement Module (BEM) (Guan et al., 2022) is integrated into the CLAHE algorithm to focus the network’s attention on contour representation. This module leverages both high-level and low-level features to isolate boundary details and eliminate irrelevant information. Its operational mechanism is depicted in Fig. 1. The image information input into the image preprocessing algorithm enters both the CLAHE algorithm channel and the boundary detection enhancement channel. The CLAHE algorithm channel adjusts image contrast, while the front and rear edge enhancement channel concentrates on extracting and intensifying boundary information within the image. This process not only enhances the boundary information between the foreground and background but also accentuates the shape details of the target object. Subsequently, the outputs from both channels are merged, effectively enhancing the underwater image alongside the object’s shape and boundary details concurrently.

Figure 1 The working mechanism of the CLAHE underwater image enhancement algorithm with foreground and background boundary enhancement module.

The structure of yolov8n model and improvement strategy

YOLOv8n

YOLOv8, the most recent advancement (Wang et al., 2023) in the YOLO series, is capable of performing both detection and classification tasks. This SOTA model builds upon previous iterations, incorporating novel features and enhancements. Notable innovations include a redesigned backbone network, an anchor-free head, and an updated loss function. This study utilizes the YOLOv8n variant as the foundational model.

As depicted in Fig. 2, the YOLOv8n architecture comprises four main components: Input, Backbone, Neck, and Head (Oh & Lim, 2023). The Input segment employs Mosaic for data augmentation, which is deactivated during the final 10 epochs. The anchor-free mechanism predicts the object’s center directly, eliminating the need for anchor box offset predictions and thereby streamlining the non-maximum suppression (NMS) process.

Figure 2 Structure of YOLOv8n network.

The Backbone serves as the primary feature extraction module, encompassing Conv, C2f, and SPPF modules. The Conv module applies convolution, BN, and SiLU activation to the input image, maintaining a similar structure to YOLOv5’s backbone. Adopting the CSP concept, the C3 module is substituted with a C2f structure to enhance gradient flow, with channel numbers tailored to different model scales. Additionally, YOLOv8 retains the SPPF module from the YOLOv5 (Wu et al., 2021) design, enabling the conversion of variable-sized feature maps into fixed-size feature vectors.

The Neck’s primary role is to amalgamate multi-scale features and construct a feature pyramid, utilizing the PANet architecture (Liu et al., 2023). This integral structure comprises two main components: the feature pyramid network FPN and the path aggregation network PAN. The synergy between FPN and PAN enables comprehensive integration of bidirectional information flow within the network, enhancing its detection capabilities.

The Head segment fragment extracts the classification and positioning details of objects of different sizes from feature maps of different sizes. In YOLOv8n, the Head has been updated to the prevalent decoupled head structure, which segregates the classification and detection functions and transitions from an Anchor-Based to an Anchor-Free approach.

Deformable convolutional network (DCN) module

Given the flexible nature and variable shapes of marine biological objects, object detection algorithms need to enhanced capabilities to model geometric transformations. Traditional convolution operations partition the feature map into segments matching the convolution kernel’s size, with each segment’s position on the feature map being static. This approach is less effective for objects undergoing complex deformations. As illustrated in Fig. 3A, the fixed size and shape of a 3 × 3 convolution kernel limit its capacity to model geometric transformations during convolution. To address this, researchers like Dai et al. (2017) introduced an innovative concept of integrating an offset to the standard convolution kernel, as detailed in Fig. 4, showcasing the principles of deformable convolution operations. Figures 3B–3D depict how each element of the deformable convolution kernel is adjusted by a specific offset, altering the sampling positions (Nguyen et al., 2023) and thereby augmenting the convolutional neural network’s capability to model geometric transformations and improving the model’s proficiency in detecting flexible marine objects.

Figure 3 (A–D) Sampling point location map of deformable convolution.

Figure 4 Principles of deformable convolution operation.

Traditional two-dimensional convolution first uses the sampling grid R to sample on the input feature map x (Li & Liu, 2023), and then uses the weight w to weight the sampled values. The sampling grid determines the size of receptive field. For example, the sampling grid of a regular two-dimensional convolution of size 3 × 3 can be expressed as:

(6) R=[(−1,−1),(−1,0),...,(0,1),(0,1)].

For each position p0 in the output feature map y, there is

(7) y(p0)=∑pn(pn)⋅x(p0+pn)⋅

In Formula (7), pn denotes the enumeration of positions in R. In deformable convolution, the regular grid R is offset by {δp0=n=1,...,N}, in which N=|R|. Formula (7) then becomes the following form:

(8) y(p0)=∑pn(pn)⋅x(p0+pn+δp0)⋅

This adjustment causes the sampling points to assume irregular and shifted positions.

In deformable convolution, δp0 is determined by convolving on x. The process of learning δp0 and generating the convolution kernel in deformable convolution is depicted in Fig. 4. Furthermore, the incorporation of multiple deformable convolutional layers significantly influences the receptive field and composite deformation, as illustrated in Fig. 5. Figure 5A employs standard convolution, whereas Fig. 5B utilizes deformable convolution. Both figures share identical top-level feature maps and sampling points. However, through successive convolutional layers, it is evident that deformable convolution allows for the adaptive adjustment of sampling points in response to the target’s scale and shape.

Figure 5 (A–D) A comparison of receptive field and positions of sampling points of regular convolution and deformable convolution operation.

Convolutional neural networks that incorporate deformable convolution are referred to as deformable convolutional neural networks (DCNN). From the discussed content, it is clear that a DCNN model equipped with deformable convolution can effectively modify the receptive field, enrich the feature map’s useful feature information, and enhance the network model’s capacity to model the geometric transformation of the detected target based on its size and shape.

Compared to traditional convolution, deformable convolution introduces additional parameters and greater flexibility to the network model, but this can complicate the convergence of the model. Improper design or placement of the deformable convolution module within the network might lead to training collapse.

To mitigate the risk of network collapse during training, this study draws on the classic ResNet (Shafiq & Gu, 2022) structure to inform the design of the deformable module unit, including its stacking method and placement. A skip connection is employed between two deformable convolution layers to enhance network convergence, as depicted in Fig. 6, which illustrates the deformable convolution basic module structure utilized in this experiment. CBL module refers to “convolutional block laver”, which is a basic convolutional block composed of a series of convolutional layers, batch normalization layers and activation function layers.

Figure 6 The structure of deformable convolution module.

Consequently, the DCN module is more adept at managing non-rigid deformations and positional variations of underwater targets, enhancing detection accuracy. Through learning deformable convolution parameters, the model dynamically adjusts the convolution kernel’s sampling positions based on the target’s specific shape and location, enabling more precise target characterization. This enhancement bolsters the underwater target detection model’s performance, increasing its adaptability to the complexities and variability of underwater environments.

RepBi-PAN neck network

The Neck network plays a crucial role in applying transformation and concatenation operations to the feature map, enriching and diversifying its feature information while enhancing its robustness. In YOLOv8n, this network incorporates the PANet architecture, which introduces an additional bottom-up path to FPN, facilitating quicker signal transmission from low-level to high-level features and vice versa, thus enhancing the propagation of precise low-level feature signals. Numerous scholars have developed innovative neck network designs based on FPN and PAN, with RepBi-PAN standing out as a notable example.

RepBi-PAN features a Bi-directional Concatenation (BiC) module, depicted in Fig. 7. This module amalgamates feature maps from three adjacent depths by applying 1 × 1 convolutions for dimensionality reduction on feature maps of identical scale and depth. For larger-scale feature maps from shallower layers, it employs 1 × 1 convolution for dimensionality reduction followed by a 3 × 3 convolution with a stride of 2 for down-sampling. Conversely, for smaller-scale feature maps from deeper layers, it uses a 2 × 2 transposed convolution followed by up-sampling. Afterward, it concatenates the three feature maps and applies another round of 1 × 1 convolution for dimensionality reduction. Through this process, the BiC structure assimilates feature information across adjacent layers, facilitating cross-layer and cross-channel feature fusion, enhancing the interaction between deep and shallow layers, and significantly improving the neck network’s feature fusion capability. Furthermore, this module aids in preserving precise positioning signals, which is particularly crucial for accurately locating smaller objects.

Figure 7 The structure of BiC.

In addition to the BiC structure, RepBi-PAN enhances the SSP structure (Tang et al., 2023) by streamlining the SPPF block into a version akin to CSP, termed the SlimCSPSPPF block. Figure 8 illustrates the refined SlimCSPSPPF structure. Upon receiving the deepest layer’s feature map, SlimCSPSPPF bifurcates it into two pathways. One pathway engages in multi-scale transformation of the feature map through compound convolution and spatial pyramid pooling, while the other employs a skip connection to preserve SlimCSPSPPF’s integrity. The SlimCSPSPPF structure surpasses the original SPPF in terms of feature representation, efficiency, and processing speed.

Figure 8 The structure of SlimCSPSPPF.

RepBi-PAN integrates the BiC structure to link convolutional layers across four different depths at the Backbone’s second and third levels of convolutional depth and implements the SlimCSPSPPF structure at the deepest convolutional layer. By refining the original YOLOv8n’s PANet structure, RepBi-PAN substantially boosts the model’s capacity to amalgamate multi-scale features pertinent to flexible marine organisms, thereby enhancing the detection accuracy of the algorithm for these subjects.

The SimAM attention structure

In tasks involving the detection of marine life, where the target closely resembles the background, incorporating an attention mechanism prior to the YOLOv8n network’s head can significantly mitigate the impact of extraneous information. This article employs SimAM, a three-dimensional, parameter-free attention mechanism (Zhang, Liu & Jiang, 2022).

Traditional attention mechanisms, such as SE, ECA, and SOCA, are constrained by their intrinsic designs, typically refining features along either channel or spatial dimensions and generating one- or two-dimensional weights. This constrains the model’s capacity to concurrently learn weights across both channels and spatial dimensions and escalates the model’s parameter count. Unlike these mechanisms, SimAM introduces a parameter-free approach to generate full 3D weights, achieving three-dimensional weight derivation without increasing the original network’s parameter load. Figure 9 compares the operational principles of the SimAM attention structure with those of channel and spatial attention structures.

Figure 9 (A–C) The comparison of the principles of SimAM attention structure, channel attention structure and spatial attention structure.

Figure 9 illustrates that channel attention generates one-dimensional weights for each channel, refining the feature map’s channel weights while treating all spatial positions within a channel uniformly (Zhang, Li & Zhang, 2023). Spatial attention generates two-dimensional spatial weights, refines the weights of different positions within the same channel of the feature map, and treats all channels equally. Conversely, the 3D weights generated by SimAM can refine different positions across various channels. Hence, it surpasses both channel attention and spatial attention. The mechanism through which the SimAM attention structure produces 3D weights is detailed below.

The formula for calculating 3D weights is presented as:

(9) X˙=sigmoid(1E)⊙X.

X is the input feature, and the sigmoid function is applied to constrain any excessive values in E. E represents the energy function on each channel. The calculation process of E is

(10) E=4(σ2+λ)2(t−μ)2+2σ2+2λ.

In Formula (10), t denotes the value of the input feature, t∈X. The lower the energy E is lower, greater the difference between the target neuron t and other neurons xi in the corresponding dimension, thereby increasing the relative importance. Consequently, the significance of the neuron can also be denoted by 1/E. λ is the constant 1e−4, μ and σ2 denote the mean and variance of each channel in X respectively, with the computation process outlined as follows:

(11) μ=1M∑i=1Mxi

(12) σ2=1M∑i=1M(xi−μ)2.

M denotes the number of neurons, which equals the product of the width ( W) and the height ( H) of the feature map. xi represents all the neurons in the dimension corresponding to the target neuron. By manipulating these neurons, the network can assign greater weights to those containing crucial information, thereby enhancing detection and positioning accuracy without the need for additional network parameters.

It is considered that the detection network of flexible marine biological targets is easily affected by background interference similar to the target, incorporating the SimAM attention mechanism into the detection network allows for a comprehensive and efficient evaluation of feature weights and the fusion of target information across multiple dimensions. This process not only amplifies the information pertaining to flexible marine biological targets but also diminishes the influence of natural background information, ensuring a focused target detection. Furthermore, the SimAM attention structure dose not introduce extra network parameter calculations, thus minimally affecting the model’s running speed.

The WIoU loss function

In current detection algorithms, researchers designing the bounding box loss function often presume that the training dataset’s bounding box annotations are of high quality, focusing on improving the bounding box regression function’s fitting capabilities. However, low-quality bounding box annotations are prevalent in many datasets. Enhancing the fitting ability of the bounding box regression function for these low-quality images can adversely affect the model’s target positioning performance. The training data in this study, annotated by the authors, inevitably include such low-quality annotations, which can diminish the model’s generalization capability.

The baseline model of YOLOv8 employs the CIoU loss function, which considers the overlapping area, center point distance, and aspect ratio. However, the aspect ratio is represented as a relative value, introducing a degree of ambiguity. Additionally, it fails to address the balance issue between challenging and straightforward samples, potentially leading to slow convergence and reduced prediction accuracy. Given the presence of low-quality samples in our dataset, factors like aspect ratio and distance can intensify penalties for these samples, undermining the model’s generalization capabilities. When there is significant overlap between the target and anchor boxes, reducing the emphasis on geometric factors can enhance model generalization. Consequently, this study adopts the WIoU loss function over the CIoU loss function. The WIoU loss function more effectively addresses sample imbalance, improving the model’s convergence rate.

WIoU employs a dynamic, non-monotonic focusing mechanism to assess the anchor box’s quality and leverages gradient gain to bolster the efficacy of high-quality anchor boxes while mitigating the effects of detrimental gradients (Chen et al., 2023b), thereby enhancing the algorithm’s overall performance. WIoU introduces a dual-layer attention mechanism to avert slow convergence, augment convergence accuracy, and bolster model generalization. It is posited that the corresponding position of (x, y) in the target frame is (xgt,ygt), RWIoU denotes the loss of high-quality anchor boxes. The WIoUv1 formula is presented as:

(13) LWIoUv1=RWIoULIoU

In which

(14) RWIoU=exp((x−xgt)2+(y−ygt)2((wc)2+(hc)2)∗)

To circumvent the generation of substantial harmful gradients by low-quality samples, WIoUv3 is devised utilizing β and WIoUv1. The formula is delineated as follows:

(15) LWIoUv3=rLWIoUv1.

In which

(16) r=βαβ−δ,β=LIoU∗LIoU∈[0,+∞).

This study employs the WIoUv3 loss function for experimental analyses. This function incorporates a dynamic, non-monotonic focusing mechanism (Wang et al., 2023). It dynamically computes the “outlier degree” to serve as a metric for evaluating the quality of anchor boxes and implements a gradient gain allocation strategy. A lower outlier degree indicates a higher-quality anchor box, to which a reduced gradient gain is allocated, refocusing the bounding box on anchor boxes of average quality. Conversely, assigning diminished gradient gains to anchor boxes with higher outlier degrees effectively mitigates the impact of low-quality examples by limiting their potential to generate large, detrimental gradients. Furthermore, the calculation of the outlier degree is dynamic throughout the training process, enabling WIoU to continually adapt its gradient gain distribution strategy to optimize performance in the prevailing context.

As demonstrated in Fig. 10, Wise-IoU can reduce the influence of these low-quality labels and achieve better prediction when the labeling frames of jellyfish and holothurian in training data are quite different from the real ones. The dataset for this study was developed through the authors’ independent collection and annotation of images, encompassing over 7,000 annotated boxes set against a complex seabed environment background.

Figure 10 Low-quality bounding box annotations in the training and ideal predictions of Wise-IoU.

Given the inevitability of missing and low-quality annotations, incorporating WIoU into the detection algorithm significantly reduces the adverse effects of substandard training data, boosts the bounding box regression function’s adaptability, and enhances the algorithm’s performance.

The overall framework of marine flexible biological target detection algorithm based on YOLOv8n

Following the evaluation of the target dataset and application context, enhancements were made to the original detection algorithm. Figure 11 illustrates the comprehensive architecture of the refined flexible marine biological object detection algorithm network, based on YOLOv8n. This architecture combines the original YOLOv8 model with the improvements mentioned above.

Figure 11 The overall structure of the improved algorithm model.

Experiment research and result analysis

Configuration of the experiment

The experimental environment comprises Ubuntu 20.04 64-bit, PyCharm 2023.1.2, Pytorch 2.0.1, CUDA 11.8 (for training acceleration), and Python 3.11. The hardware setup includes an AMD Radeon 7-5700X CPU and an RTX4080 GPU. These processors are utilized for image preprocessing and model training, respectively. The enhanced algorithm’s outcomes are benchmarked against those of leading target detection algorithms using GPU-accelerated training.

Evaluating index of the model

The evaluation metrics are precision, recall, and mean average precision (mAP).

Precision: the ratio of correctly predicted positive observations to the total predicted positives.

(17) Precision=TPTP+FP.

Recall: the ratio of correctly predicted positive observations to all observations in actual class.

(18) Recall=TPTP+FN.

Average precision: By plotting recall on the x-axis and precision on the y-axis, a curve is generated. The area under this curve represents the AP.

(19) AP=∫P(R)dR.

AP represents a detailed mean accuracy, computed at intervals of 0.05 across a range from 0.5 to 0.95. Due to its complexity, the currently commonly used index is AP50, which refers to the average accuracy when IoU = 0.5. AP50 generally refers to the average detection accuracy of a single category of targets. The weighted average AP50 value of multiple categories is also called mAP50 (%) (mean average precision). mAP50 (%) offers a comprehensive assessment of the model’s detection accuracy across all categories, serving as the experimental evaluation metric in this study. The mean average precision is delineated in Formula (20), where N signifies the number of target categories, and AP represents the average accuracy of a single category target.

(20) mAP=∑i=1NAPiN.

Dataset collection

a) Dataset. To ensure the dataset’s representations of biological object postures, locations, and scenarios accurately reflect real-world conditions and closely align with the actual data distribution, this study sourced extensive footage of authentic oceanic environments from documentaries, employing frame-by-frame annotation to develop a dataset of flexible marine biological entities. The utilized documentary footage is publicly available, making it suitable for this research’s dataset needs. Figure 12 displays the distribution of target categories within the dataset, which comprises a total of 6,749 images featuring prevalent flexible marine species, such as octopus, holothurian, starfish, and jellyfish. Images were annotated using Labelimg. The dataset is divided into a training set with 5,341 images and a validation set consisting of 1,408 images. It encompasses a diverse array of images captured across various seasons, showcasing the organisms’ body postures during different activities and habitats, under varying light conditions and environmental settings. Figure 13 presents a selection of images from the dataset.

Figure 12 The histogram of categories of flexible marine biological objects.

Figure 13 Some images in the dataset.

b) Dataset image enhancement. Given that the subjects of this dataset are flexible biological entities inclined to camouflage and blend into their surroundings, they often appear in environments closely resembling themselves. Consequently, this study introduces targeted enhancements to the standard CLAHE underwater image enhancement algorithm, incorporating boundary detection into the enhancement module to accentuate the distinction between foreground and background while adjusting contrast. As illustrated in Fig. 14, the contrast in images processed with HE and CLAHE has notably improved, with a clear enhancement in detail visibility. However, some HE-processed images exhibit color shift issues, particularly noticeable in the second image. DCP-processed images show reduced contrast and intensified blue-green color shifts, alongside diminished overall brightness. While RGHS processing improves image brightness, it can lead to distortion and a reddish color tint. In contrast, images enhanced with the modified CLAHE algorithm display enhanced detail clarity without significant color shifts or localized over-enhancement. Overall, the image quality significantly benefits from the enhancements applied by the refined CLAHE algorithm.

Figure 14 Effects of different image processing algorithms.

To evaluate the impact of various image enhancement algorithms on model detection outcomes, this study utilized images processed by different algorithms for model training and testing. The experimental results, presented in Table 1.

Table 1 Performance index of different models comparison table.

Methods	AP50 (%)	mAP50 (%)	
Holothurian	Starfish	Jellyfish	Octopus	
Original image	66.5	77.9	80.5	78.5	75.9	
DCP	67.9	78.1	80.3	79.7	76.2	
CLAHE	67.7	77.7	82.2	79.2	76.7	
HE	68.1	77.2	80.4	79.8	76.4	
RGHS	69.0	78.1	79.9	80.5	76.5	
Improved CLAHE	68.5	75.5	81.1	81.2	77.1	

The experimental results reveal that, relative to the original image, model detection accuracy has improved following preprocessing with CLAHE, RGHS, HE, and DCP. Notably, there was a significant enhancement in the detection quality of octopus. The mAP50 (%) for the enhanced CLAHE algorithm increased by 1.2% (relative to the original image), followed by HE at 0.5%, RGHS at 0.6%, DCP at 0.3%, and standard CLAHE at 0.8%. These findings convincingly demonstrate the effectiveness of the enhanced CLAHE algorithm in augmenting underwater images, prompting its selection for future experiments in this study.

Model training

Table 2 outlines the model training parameter settings. The dataset is partitioned into a training set comprising 5,341 images and a validation set with 1,408 images. Moreover, to mitigate overfitting during training, a label smoothing technique is employed.

Table 2 Model training parameters.

Name of parameter	Meaning	Setting	
Pretraining weight	Initial weights of the model	YOLOv8n.pt	
Optimizer	Parameter updating algorithm of the model	Adam	
Image-size	Input image size of the model	640 × 640	
lr0	Initial learning rate	0.01	
Batch size	Batch size	16	
Epoch	Training times of all training sets	200	

Experiment results and analysis

Effectiveness verification of the improved algorithm

To assess the efficacy of the improved algorithm module, this study employs the original YOLOv8n model as the baseline model and mAP50 (%) as the evaluation metric. Ablation studies were performed using various combinations of enhancement modules. Compared to the baseline YOLOv8n model, the following five enhancements each contributed to an increase in model detection accuracy, as detailed in Table 3. Post-processing with the enhanced CLAHE algorithm resulted in a 1.2% increase in mAP50 (%). Substituting the original C2f module with deformable convolution (DCN) led to a 1.3% rise in mAP50 (%). Incorporation of the RepBi-PAN network model into the neck improved feature fusion, while the introduction of the SimAM attention module augmented the algorithm’s feature processing capabilities. Subsequently, adopting the WIoU bounding box loss function in place of the original CIoU further elevated mAP50 (%). Of all modifications, the DCN module exhibited the most substantial enhancement, showing the most pronounced effect. Collectively, these modifications elevated the YOLOv8n-Improve’s mAP50 (%) by 4.6% compared to the original YOLOv8n, indicating significant improvement.

Table 3 Ablation experiment result.

Model	CLAHE	DCN	RepBi-PAN	SimAM	WIoU	Recall (%)	Precision (%)	mAP50 (%)	
YOLOv8n						70.7	82.5	75.9	
√					69.8	84.8	77.1	
√	√				71.1	86.6	78.4	
√	√	√			70.8	87.3	79.1	
√	√	√	√		72.2	87.9	80.0	
√	√	√	√	√	71.9	89.8	80.5	

Figure 15 presents the P-R diagram, illustrating the sequential addition of improvement methods. The detection accuracy for sea cucumbers, starfish, octopus, and jellyfish has seen overall enhancements through these methods. Specifically, the detection accuracy improvements are as follows: sea cucumbers by 8%, octopus by 6.1%, starfish by 3.1%, and jellyfish by 1.3%. Consequently, the final detection accuracies are 74.5% for sea cucumbers, 84.6% for octopus, 81.0% for starfish, and 81.8% for jellyfish, indicating substantial improvement.

Figure 15 The P-R diagram in which the original YOLOv8 and the improving methods is added.

Figure 16 illustrates that YOLOv8n-Improve converges more rapidly, and its loss function values are lower compared to YOLOv8n. The integration of the DCN module enhances feature extraction efficiency, while the inclusion of RepBi-PAN ensures more comprehensive feature fusion across layers, thereby improving bounding box regression accuracy and reducing loss values. In Fig. 17, YOLOv8n-Improve exhibits superior precision and recall relative to YOLOv8n. This improvement is attributed to the SimAM attention module’s robust feature information processing capabilities, which effectively address the challenge of feature information overload due to the diversity of flexible biological objects’ features. It emphasizes the crucial features of marine biological entities while minimizing background noise. Furthermore, the adoption of the WIoU bounding box loss function mitigates the adverse effects of low-quality annotations in the dataset.

Figure 16 Curve of loss functions of YOLOv8n and YOLOv8-Improve.

Figure 17 Curve of precision and recall of YOLOv8n and YOLOv8-Improve.

To visually assess the impact of the improved algorithm on the model’s detection performance, the detection confusion matrices of the baseline algorithm and the YOLOv8-Improve algorithm are compared, as illustrated in Fig. 18.

Figure 18 The YOLOv8-Improve algorithm is used to process the contrast graph of the confusion matrix.

Figure 18 demonstrates that, following the application of the YOLOv8n-Improve algorithm, the recall rates for jellyfish, sea cucumbers, and octopus have increased, with the exception of starfish. Additionally, the missed detection rates for all four flexible biological targets have decreased to some extent. Both Table 3 and Fig. 18 corroborate the efficacy of the proposed algorithm.

Comparison of different models’ detection results

To further assess the algorithm’s effectiveness, this study compares the performance of YOLOv8-Improve with other leading object detection algorithms using a dataset of flexible marine biological entities. The comparisons, detailed in Table 4, show that YOLOv8n-Improve offers comparable accuracy to the newly proposed Compared with the newly-proposed RT-DETR (Lv, Xu & Zhao, 2023) and the target detection algorithm for Marine environment design in literature (Lei, Tang & Li, 2022), they have similar accuracy, but YOLOv8-Improve has faster detection speed. Against the two-stage algorithm Faster-RCNN, YOLOv8n-Improve enhances the mAP50 (%) by 5.4%, with significant reductions in single-image detection time, Weights (M), and GFLOPs. The SSD algorithm underperforms on this dataset, achieving the lowest detection accuracy of 71.5% and subpar results in other metrics. Additionally, YOLOv8n-Improve is benchmarked against lighter YOLO series networks like YOLOv5n, YOLOv7-tiny, and YOLOv8n. Table 4 reveals that YOLOv8n-Improve’s mAP50 (%) is 80.5%, which is 4.6% higher than the unenhanced YOLOv8n, with minimal impact on detection speed. The model’s detection rate of 109.8 fps surpasses the 60 fps threshold, suitable for real-time scene detection. It shows slight improvements in Weights (M) and GFLOPs over YOLOv7-tiny and YOLOv5s, with mAP50 (%) gains of 3.4% and 3.0%, respectively. Furthermore, its single-image detection time of 9.1 ms is more efficient than YOLOv7-tiny’s 14.8 ms and YOLOv5s’s 12.1 ms. In summary, YOLOv8n-Improve demonstrates commendable detection performance.

Table 4 Comparison of different models’ detection results.

Model	mAP50 (%)	Weights (M)	GFLOPs	Single image detection time (ms)	
SSD	71.5	97.6	273.4	22.1	
YOLOv5s	77.5	14.4	16.0	12.1	
YOLOv7-tiny	77.1	12.5	13.2	14.8	
Faster-RCNN	75.1	109.5	370.2	25.2	
YOLOv8n	75.9	6.3	8.2	7.2	
RT-DETR	80.3	32.8	110.2	15.8	
Literatures (Chen et al., 2023b)	80.2	16.6	20.5	26.8	
YOLOv8n-Improve	80.5	7.6	9.1	9.1	

To more vividly illustrate the algorithm’s detection capabilities, this study compares the detection outcomes of Faster-RCNN, YOLOv8n, and the developed YOLOv8n-Improve for various marine biological entities. Figure 19, which displays the bounding boxes and confidence levels for detections of octopus, jellyfish, starfish, and sea cucumber, shows that YOLOv8n-Improve outperforms the other models in terms of bounding box accuracy and confidence. Specifically, YOLOv8n occasionally misses smaller objects like starfish and sea cucumbers. Moreover, in scenes with multiple objects or significant object deformation, Faster-RCNN and YOLOv8n sometimes produce false detections, evident in the jellyfish and octopus instances. These observations affirm the superior detection efficacy of YOLOv8n-Improve on flexible marine organisms.

Figure 19 Visual comparison of different models’ detection results.

Conclusions

This study addresses the challenge of detecting flexible marine biological objects, which are characterized by their soft bodies, ease of deformation, large deformation scales, and varied postures. Initially, a dataset specific to these organisms was created. Subsequently, the CLAHE underwater image enhancement algorithm was employed, augmented with a module to enhance foreground and background boundaries for image processing. Considering the unique traits of these organisms, the YOLOv8 algorithm was selected as the baseline for object detection experiments, leading to the development of an enhanced YOLOv8-based detection algorithm. This enhancement includes a deformable convolution module to improve the feature extraction network’s ability to model the geometric variations of the organisms. Additionally, the RepBi-PAN network structure was incorporated to bolster the feature aggregation capability of the algorithm, while the SimAM attention mechanism was integrated to refine feature processing and reduce hardware demands. The introduction of the WIoU bounding box loss function aims to mitigate the effects of poor-quality labels. Although the proposed algorithm exhibits a slight decrease in detection speed compared to the baseline, it maintains adequate real-time performance and achieves a 4.6% increase in detection accuracy, demonstrating its effectiveness for detecting flexible marine biological entities.

Additional Information and Declarations

Competing Interests

Author Contributions

Data Availability

The authors declare that they have no competing interests.

Yu Tian conceived and designed the experiments, performed the experiments, analyzed the data, performed the computation work, prepared figures and/or tables, authored or reviewed drafts of the article, and approved the final draft.

Yanwen Liu conceived and designed the experiments, authored or reviewed drafts of the article, and approved the final draft.

Baohang Lin analyzed the data, prepared figures and/or tables, and approved the final draft.

Peng Li conceived and designed the experiments, performed the experiments, analyzed the data, authored or reviewed drafts of the article, and approved the final draft.

The following information was supplied regarding data availability:

The data is available at figshare: Tian, Yu (2024). Dataset.zip. figshare. Dataset. https://doi.org/10.6084/m9.figshare.24447487.v1.

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
