# Peer review of "Research on marine flexible biological target detection based on improved YOLOv8 algorithm"

_PeerJ Computer Science, doi:10.7717/peerj-cs.2271_

## Round 0.1 · original submission · Major Revisions

Both referees make clear requests about the extra work and discussion that they would like to see. Please consider their comments carefully in making your new version, and note that it will be sent out to referees again.

**Language Note:** PeerJ staff have identified that the English language needs to be improved. When you prepare your next revision, please either (i) have a colleague who is proficient in English and familiar with the subject matter review your manuscript, or (ii) contact a professional editing service to review your manuscript. PeerJ can provide language editing services - you can contact us at [email protected] for pricing (be sure to provide your manuscript number and title). – PeerJ Staff

Reviewer 1 ·

Basic reporting

The manuscript presents a novel approach to improving the detection of marine flexible biological targets by utilizing an enhanced version of the YOLOv8 algorithm integrated with various innovative techniques. The authors aim to tackle the challenges posed by underwater image processing and object deformation, contributing to advancements in marine detection systems. However, the paper requires substantial revisions to reinforce its scientific rigor and clarity.
My concerns:
1. Introduction and Background:
Provide a thorough exposition of the need for detecting marine flexible biological targets, including its relevance to oceanic applications and correlation to the current body of literature. This would enhance understanding of the study's significance.
2. Literature Review:
Bolster the literature review with additional, current sources pertinent to marine flexible biological target detection. Ensure that these references substantiate the development of your proposed algorithm. To enrich the analysis of the research topic on biological target detection and provide a more in-depth exploration of the Deep learning methodology, it is imperative for the manuscript to undertake a critical review of pertinent literature. The following papers should be meticulously examined and discussed within the context of your literature review:

* Chen Y, Yang X H, Wei Z, et al. Generative adversarial networks in medical image augmentation: A review[J]. Computers in Biology and Medicine, 2022, 144: 105382.
* Guan Q, Chen Y, Wei Z, et al. Medical image augmentation for lesion detection using a texture-constrained multichannel progressive GAN[J]. Computers in Biology and Medicine, 2022, 145: 105444.
* Wu W, Liu H, Li L, et al. "Application of local fully Convolutional Neural Network combined with YOLO v5 algorithm in small target detection of remote sensing image". PloS one, 2021, 16(10): e0259283.
* Ju M, Luo H, Wang Z, et al. "The application of improved YOLO V3 in multi-scale target detection". Applied Sciences, 2019, 9(18): 3775.
* Jiang P, Ergu D, Liu F, et al. "A Review of Yolo algorithm developments". Procedia Computer Science, 2022, 199: 1066-1073.

Incorporate an analysis of these articles to highlight their relevance to your work, discuss how they align or contrast with your proposed improvements, and demonstrate the novelty of your approach within the broader context of existing YOLO variants and applications.

Experimental design

3. Methodology – Dataset Description:
Elaborate on the dataset details concerning your marine flexible biological targets, detailing the dataset's scale, compilation method, and any inherent data biases or limitations.
4. CLAHE Algorithm and Boundary Detection Enhancement Module:
Clarify how the CLAHE algorithm with the boundary detection enhancement module is executed to augment underwater imagery and delineate between foreground and background more effectively.
5. DCN Network Module Justification:
Thoroughly justify the decision to supplant the C2f module with a DCN network module within the YOLOv8n framework, highlighting how it enhances geometric transformation modeling and key area focus.
6. RepBi-PAN Structure Explanation:
Provide comprehensive insights into the RepBi-PAN structure employed in the neck network, elucidating how it amalgamates and accentuates features crucial for identifying flexible biological targets.
7. SimAM Attention Structure Role:
Delve deeper into the SimAM attention structure's contribution, explicating its role in fortifying the model's feature information processing and its effect on the accuracy of target detection.
8. WIoU (Wise-IoU) Loss Function:
Explain the WIoU loss function's implementation in addressing low-quality labels within the dataset and how it refines anchor box evaluation metrics.

Validity of the findings

9. Simulation Experiment Results:
Present the simulation experiment outcomes meticulously, employing specific metrics and statistical evidence to substantiate the claim of significant accuracy enhancement in target detection.

Additional comments

10. Language and Clarity:
Review the manuscript to correct grammatical inaccuracies and refine the writing for succinctness and clarity. A professional editorial service is recommended to elevate the language quality to publication standards.

Reviewer 2 ·

Basic reporting

Marine flexible biological targets often exhibit deformable and varying scales, making it challenging to design an algorithm that effectively captures these characteristics.
My observations is that the paper addresses this by introducing a deformable convolution module and a RepBi-PAN neck network structure to enhance the algorithm's ability to model geometric transformations and aggregate key features of flexible biological targets with varying scales.

Simlarly, the similarity between the foreground and background in underwater images can lead to issues in differentiating marine flexible biological targets from their surroundings. It is appreciable that the authors tackle this challenge by proposing a CLAHE algorithm with a boundary detection enhancement module for underwater image enhancement. This aims to enhance the boundary between foreground and background, improving target distinctiveness.
False Detections and Multiple Objects:

The Images with multiple detection objects and large deformations may result in false detections. How this issue can be minimized?

Experimental design

The negative impact of low-quality labels in the dataset can affect the performance of the algorithm. Authors are advised to justify it explicitly.
The manuscript mentions the use of AMD Radeon 7-5700X CPU and RTX4080 GPU for training, but lacks details on how these hardware components were effectively utilized during the experiments. A detailed explanation or analysis of the hardware efficiency in terms of parallel processing, memory usage, and overall training acceleration would enhance the completeness of the study.

Validity of the findings

Authors are advised to the issue of the introduced algorithm which slightly reduces the detection speed compared to the baseline algorithm.

Additional comments

The dataset collection process is briefly described, mentioning real ocean scenarios, but lacks information on the diversity of locations, lighting conditions, and seasons represented in the dataset. Providing details on how the dataset captures the variations in marine environments would strengthen the generalization claims of the proposed algorithm.

While the paper discusses the effectiveness of the improved algorithm, it lacks comprehensive robustness testing against diverse conditions, such as varying underwater visibility, background complexity, and target occlusions. Including results and insights from experiments conducted under different challenging scenarios will strengthen the algorithm's claim for real-world applicability.

The manuscript compares the proposed algorithm with YOLOv8n and YOLO series, but lacks a comparison with other state-of-the-art object detection algorithms, especially those designed for marine environments. Including a comparative analysis with contemporary algorithms tailored for underwater object detection will provide a more comprehensive understanding of the proposed method's strengths and weaknesses.

While the manuscript reports improvements in detection accuracy, it lacks a detailed discussion on the choice of evaluation metrics and their appropriateness for underwater object detection. Providing insights into why specific metrics, such as mAP50, were chosen and their relevance to the characteristics of marine flexible biological targets would enhance the paper's scientific rigor.

The paper mentions collecting videos from documentaries, but it lacks information on ethical considerations, such as permissions for data usage, adherence to privacy guidelines, and potential environmental impact. Including a brief section on ethical considerations in dataset collection would demonstrate the authors' commitment to responsible research practices.

The manuscript briefly touches upon the reduction in detection speed without providing detailed analysis or comparisons with real-time requirements. Including a dedicated section on the algorithm's real-time performance, discussing frame-per-second rates, and addressing any trade-offs made for achieving real-time capabilities will strengthen the practical utility of the proposed algorithm.

---

## Round 0.2 · Minor Revisions

I apologise for the delays in responding, which were due to some refereeing delays. You will see that a new referee has looked at the paper. While they have some suggestions, the only important one is "The literature review should include more recent advancements in underwater object detection." Please consider if there are relevant recent references, and submit a final version as soon as possible.

Reviewer 3 ·

Basic reporting

The manuscript is mostly clear, but some areas require further polishing for clarity and professionalism. The literature review should include more recent advancements in underwater object detection. The article follows a standard structure, but some figures could be simplified or annotated for better readability. The submission is self-contained with all relevant results.

Experimental design

The research question is well defined and addresses a significant gap. The investigation is rigorous and meets high technical standards. Methods are described in detail, though additional visual examples of preprocessing steps would enhance understanding.

Validity of the findings

The study provides a meaningful contribution to the literature. The data are robust, statistically sound, and controlled. Conclusions are well stated and linked to the original research question, supported by the results.

Additional comments

The authors have addressed many previous comments effectively but could include more information on dataset diversity, conduct additional ablation studies, and substantiate the algorithm's robustness under challenging conditions.

---

## Round 0.3 · accepted · Accept

Thank you for addressing the comments of the referee so speedily. I confirm that the content of the manuscript is now ready for publication, based on my reading of your paper. the referee's comments, and your rebuttal.